# Primary Drivers of Willingness to Continue to Participate in Community-Based Health Screening for Chronic Diseases

**DOI:** 10.3390/ijerph16091645

**Published:** 2019-05-11

**Authors:** Shih-Ying Chien, Ming-Chuen Chuang, I-Ping Chen, Peter H. Yu

**Affiliations:** 1Institute of Applied Arts, National Chiao Tung University, Hsinchu 30010, Taiwan; cming@faculty.nctu.edu.tw (M.-C.C.); iping@faculty.nctu.edu.tw (I.-P.C.); 2Institute of East Asian Studies, University of California Berkeley, Berkeley, CA 94704, USA; 3A&M Rural &Community Health Institute and Department of Primary Care Medicine, Texas A&M University, College Station, TX 77845, USA; peteryu@tamhsc.edu

**Keywords:** chronic diseases, community-based health screening, participation rate, sociodemographic, willingness to continue to participate

## Abstract

Background: As the average age of the population continues to rise in the 21st century, chronic illnesses have become the most prominent threats to human health. Research has shown that early screenings for chronic diseases are an effective way of lowering incidence and mortality rates. However, low participation rates for health screening is one of the main challenges for preventive medicine. The objective of this study was to determine the primary drivers which: (1) first motivate people to participate in community-based health screening for chronic diseases; and (2) increase their willingness to continue to participate. Methods: A total of 440 individuals between 30 and 75 years of age were invited to undergo a health screening and then complete an interview questionnaire. Screenings and interviews were conducted in four regions in northern Taiwan. The questionnaire was separated into three sections, which explored sociodemographic differences, drivers of willingness to participate, and willingness to continue to participate respectively. Raw data was analyzed using the statistical software package SPSS (SPSS Inc., Chicago, IL, USA). Main Outcome Measures: Effects of sociodemographic factors on health screening participation rate, drivers of willingness to participate, and willingness to continue to participate. Results: Seventy-three percent of participants responded that they would be willing to continue to join in future health screenings. Notably, elderly people and married people were respectively more likely to participate in preventive health screening than were younger people and people who were single, divorced, separated, or widowed. Level of education was another key driver of willingness to participate in health screening for chronic diseases, as were the concern of relatives/friends and the provision of participation incentives. Discussion: Some of our findings, such as the key drivers of willingness to continue to participate in health screening that we identified, were different from findings of many previous studies conducted in other countries. The current study also found that a higher percentage of participants would be willing to join a similar health screening in the future if the service design is considered in advance and is well-implemented.

## 1. Introduction

According to the World Health Organization [1,2], chronic diseases and cancer are the leading causes of death and disability worldwide. The high rates of chronic diseases and cancer not only threaten public health and cause a decrease in quality of life, but pose serious financial burdens to individuals, families, and societies [3].

Research has consistently shown that early screening is an effective way to reduce morbidity and mortality [4], and community-based health screening has been available since 1996. However, participation rates have been low in many countries [5,6,7], and researchers and service providers have tended to ignore significant drivers of declining participation rates. Furthermore, socio demographic factors such as age, marital status, and education level are often influencing the use of health services by people with chronic diseases [8,9]. The best example can be seen unemployment and low economic status have been associated with health problems. In order to deal with the socio demographic factors that affect health outcomes, research has shown that collaboration with community groups can lead to a higher participation rate, up to 90% [10].

Nonetheless, a number of factors affect patient willingness to undergo health screening. For example, some people do not undergo health screening due to a lack of awareness [11]. Indeed, health knowledge is an important causal factor of behavior [12]. It has been found that when a patient lacks knowledge, they are less likely to be engaged in health screening decisions. Other patients believe that, unless they show symptoms of a disease, they must be healthy; therefore, health screening is a waste of time [13]. In addition, another study showed that middle-age men who smoke were less interested in preventative screenings and therefore had lower health screening participation rates [2,14]. Other patients have reported that they have not undergone health screenings due to a lack of time and/or difficulty finding someone to take care of their children while they are away [15]. Moreover, some people have chosen to not participate in health screening because they are afraid of the potentially serious health problems that screenings can reveal. Still other patients have reported that participation in previous health screening programs was an uncomfortable experience [16]. For example, some women have chosen to forego breast cancer screening due to the pain they experienced during previous mammograms. On the other hand, a number of studies [17,18,19,20] have compared the differences between rural and urban residents in the utilization of medical care services. Normally rural areas indeed have lower utilization and willingness of health care services due to the education level, service needs and service provided model, especially areas having particular health risks.

Research has also identified a connection between service design (i.e., recruitment strategies) and health screening participation rates [9]. A better understanding of public attitudes towards health screening would help healthcare providers reformulate healthcare services to encourage participation in screenings and thereby increase participation rates. This is important to not only reduce the burden of disease, but because research has confirmed that higher participation rates increase the cost-effectiveness of health screenings [21]. 

Our research objectives were to: (1) identify the sociodemographic factors which influence health screening participation rates; and (2) identify the primary drivers of patient willingness to continue participate in health screening for chronic diseases, both now and in the future. Specifically, we sought to answer the following research questions:What are the relative influences of various socio-demographic factors on health screening participation rates?What were the main drivers of participation in this health screening program?How many patients will continue to take advantage of health screening services in the future?

Participants in this study received a free health screening which evaluated indicators pertaining to body mass index (body weight, body height), waist girth (circumference), urine analysis, blood pressure, cholesterol, blood glucose level and presence/absence of helicobacter pylori. Upon completing the health screening, participants were asked to fill in a survey questionnaire (administered during weekends only). All participants received physical examination report as well as compensation one month later.

## 2. Methods

This study was integrated into community chronic disease prevention and screening services, which were provided from 20 April 2017 to 24 March 2018 in northern Taiwan. Specifically, this screening service was offered in four districts: the Anle District of Keelung City, and the Ruifang, Gongliao, and Wanli Districts of New Taipei City, and districts’ sociodemographic and clinical characteristics of the participants are shown in Table 1.

A total of 428 accepted the invitation in four districts in northern Taiwan, and. All participants were between 30 and 75 years old, and were regular smokers and drinkers. Posters, invitation letters, and leaflets were used to promote this health screening, offered by the Chang Gung Memorial Hospital which is one of the top rated medical centers in Taiwan. The liver function test, lung function test, renal disease test and preliminary cancer diagnosis health screening were first conducted, the detailed examination will be provided in follow up hospital visit. In addition to health screening, questionnaire was conducted by a research team based in Chang Gung Memorial Hospital (CGMH). The questionnaire was developed based on an extensive literature review, which included the following topics: the current state of community-based health screening services, patient experiences, and service procedures. In so doing, we aimed to identify factors which influence patient willingness to continue to participate in community health screenings. Our questionnaire was divided into three sections and employed a five-point Likert-type scale. The first section gathered socio-demographic data, including information pertaining to gender, age, marital status, level of education, and self-assessment of health status. The second section was aimed at identifying critical primary drivers which convinced participants to undergo health screening. The final section sought to determine whether participants would continue undergoing similar health screenings in the future. Methods used for health screening were approved by the Chang Gung Memorial Hospital (CGMH), and the questionnaire survey was approved by the Chang Gung Medical Foundation Institutional Review Board (IRB No. 201700302B0). Please note that some responses received were less than 428 because some surveys were returned incomplete. In this study, raw data were analyzed using the statistical software package SPSS, and differences between categorical variables were compared using chi-square tests. The partial non-response rate (missing data) was less than 1%. Significant differences were determined according to 95% confidence intervals

## 3. Results

### 3.1. Response Rate

The final response rate for the survey was 97%. Participants included 141 men (33%) and 282 women (67%) (see Table 2).

### 3.2. Sociodemographic Factors and Participation Rate

As shown in Table 2, a greater number of elderly participants (over 60 years old) than younger participants (55% versus 45%) underwent health screenings. Although chronic diseases can affect people of all ages, they are more common among the elderly.

Around 73% of respondents answered that they would be willing to continue participate in future health screenings. In Table 2, participants are divided into two groups: those who were willing to continue to participate in health screenings and those who were not. A univariate analysis revealed that more than twice as many females (67%) were willing to participate in health screening again than males (33%), which indicates that women were more concerned about their health status than were men. 

In addition, compared with other participants, people who were highly educated (Bachelor’s Degree or above) and people those who were married or who lived with others were more willing to undergo health screenings. This implies that a lower education level and living alone negatively impacts awareness of chronic diseases, which in turn results in reduced utilization of health services. Furthermore, people who suffer from health anxiety tended to be afraid of severe illness.

### 3.3. Impact of Socio-Demographics Characteristics

Table 3 shows the results of a stepwise logistic regression model that analyzed which socio-demographics factors affect willingness to participate in health screening. The results in the adjusted model reveal that participants who were older than 75 (odds ratio [OR] = 0.21, 95% confidence interval [CI] = 0.07–0.65) and who were widowed or living alone (OR = 0.27, 95% CI = 0.09–0.79) had the lowest participation rates for health screening. Conversely, participants who had a college degree or a higher level of education (OR = 6.8, 95% CI = 2.86–16.2) and participants with health anxiety (OR = 1.55%, 95% CI = 1.01–2.39) were more likely to participate in health screening.

### 3.4. Reasons for Participating

Table 4 shows the key factors which drive willingness to continue to join health screening. To determine this, we asked each participant what the primary reasons were which motivated them to participate in the screening. We offered the following seven categories in terms of responses: “service design”, “friends and family”, “convenient location”, “health status”, “compensation”, and “perception of personal health”. 

Participants responded using a 5-point Likert scale (1 = “strongly disagree” to 5 = “strongly agree”). The majority of participants noted that the concern of friends and family was the key factor which motivated them to participate in health screening (OR = 0.49, 95% CI = 0.276–0.881, *p* = 0.017 *). Many other participants indicated that the compensation offered was the main factor which motivated them to participate (OR = 0.51, 95% CI = 0.321–0.807, *p* = 0.004 *). Participants indicated that the following factors influenced them as follows: service design and quality (OR = 0.83, 95% CI = 0.456–1.497), convenient location (OR = 0.856, 95% CI = 0.264–2.774), desire to better understand personal health status (OR = 0.635, 95% CI = 0.177–2.282), and a desire to maintain good health (OR = 0.713, 95% CI = 0.441–1.154).

The majority of participants (73%) indicated that they were interested in attending in a similar health screening in the future; 27% reported that they were not interested (see Table 2).

## 4. Discussion

This study sought to understand the key factors which influence willingness to continue to participate in community-based health screening. In general, participation rates were determined by sociodemographic factors and the preventive services offered.

In this study, 73% of participants were willing to undergo future health screenings. In addition, participants with a higher education level and participants who were married or who lived with others showed a greater willingness to join health screening, which was different from the result of previous studies, elderly individuals who lived alone tended to be more worried about their physical health and thus more willing to participate in health screening. In addition, women paid more attention to their personal health than men, and willingness to participate in health screening increased with age.

It is also worth noting that education level had a critical influence over willingness to participate in health screening, whereby individuals who had achieved a higher level of education were more strongly motivated to join health screening. This implies that education which provides knowledge of chronic diseases can increase willingness to participate in health screening. Moreover, most of the participants who had a lower education level were from remote districts and most of the older people interviewed regretted the difficulties which their literacy problems caused, which means that education level is a significant characteristic that influences the decision to undergo health screening. Accordingly, we postulate that there is a positive correlation between level of education and willingness to participate in health screening, and it was evident that level of education was significantly associated with the utilization of health care services

Previous investigations found that people were less willing to participate because they were not interested [22] and the willingness depended on personal beliefs [15]. In our study, 27% of participants replied that they would not be willing to continue to join future health screenings because they were not satisfied with the current service design. In addition, although our study indicated that poor service design has a relatively small influence over willingness to participate in health screening, previous studies have reported that service design significantly impacts willingness to participate [9,23], which means that it is strongly related to participants’ decision [24]. 

This implies that willingness to participate would be influenced by the recruitment process design and quality of health services. The whole concept of service system design includes the announcement of the health screening program, to the establishment of registration channels, the selection of a screening location, the health screening process and feedbacks. Those details could comprise crucial factors which affect the willingness of participants to join health screening. In the current study, the recruiting process is organized by the village representative, who also helps with service promotion and registration. Since the health screenings of chronic disease were provided free of charge and with limited number of participants each time, the local residents signed in enthusiastically.

However, the service providers pointed out that some residents felt that the registration process was unfair because the village representative retained some screening opportunities for his/her own friends and family alone. When residents feel that their rights have been encroached, their willingness to continue to join health screening are directly affected. Furthermore, many participants noted that they were motivated to participate in health screening by small incentives (e.g., compensation provided by the hospital, free medical referral services, and complimentary medical consultations). It can thus be concluded that incentives could increase health screening participation rates.

Our study also revealed that women were more likely to participate in health screening than men. These findings are consistent with many previous studies [10,25]. However, some previous research found that women expressed lower willingness to participate in health screening due to bad experiences in previous screenings [25] or other reasons, such as an inability to find help taking care of children and/or other family members.

One of the strengths of the current study was the high response rate (97%). Another strength of the current study was the fact that we distinguished between the principal factors which influenced participant willingness to continue to undergo health screening services and the incentives to drive participant’s motivation to participate the health screening services. People appeared reluctant to undergo health screening again because they felt the screening process sometimes was confusing and they spent too much time waiting in line and lack of incentives. Both types of factors should be considered in order to make it easier for participants to attend health screenings.

The primary weaknesses of the current study were the considerable age variation of participants (between 30 and 75 years old) and the fact that participants were from four different districts with varying population densities. This geographic diversity might have resulted in different views pertaining to the various factors which underlie willingness to participate in health screening services. 

## 5. Conclusions

Health screening services allow diseases to be detected at an early stage, which can decrease the cost burden associated with chronic disease and protect and promote patient health. In this study, we found that the key factors which influenced participation in community-based health screening services were the concern of family and/or friends, gifts and/or compensation, as well as the availability of free medical referral services and/or complimentary medical consultations. We also found that more than half of the participants in this study were willing to participate in future community-based health screenings. If more incentives and key factors that influence willingness to continue to participate in health screening services are considered in the design of health screening services, it is likely that even more people will be motivated to participate in community-based health screening, thereby increasing the effectiveness of this service.

## Figures and Tables

**Table 1 ijerph-16-01645-t001:** Sociodemographic and clinical characteristics of the participants.

Districts	Sociodemographic	Clinical Characteristics
Anle	Urban area (general population)	General chronic diseases (Metabolism)
Ruifang	Rural area (miners)	High risk of lung, liver and kidney diseases
Wanli	Rural area (fishermen)	High risk of liver and metabolic syndrome
Gongliao	Rural area (nearby nuclear power plant, retired fishermen)	High risk of liver, metabolic syndrome and cancers

**Table 2 ijerph-16-01645-t002:** Socio-demographic characteristics of study participants (*N* = 428).

Willingness to Continue to Join Future Health Screening					
Variable	Yes = 312 (73%)		No = 113 (27%)		*p*-Value
Age (years)	Number	%	Number	%	
30–39	27	8.7	4	3.5	<0.001 *
40–49	48	15.4	3	2.7	
50–59	80	25.6	28	24.8	
60–69	102	32.7	39	34.5	
70+	55	17.6	39	34.5	
Missing data *n* = 3	-	-	-	-	
Sex					
Female	203	65.5	79	69.9	0.393
Male	107	34.5	34	30.1	
Missing data *n* = 5	-	-	-	-	
Education level					
Illiterate	20	6.4	17	15.0	<0.001 *
Elementary	41	13.2	34	30.1	
Junior high school	30	9.6	19	16.8	
Senior high school	103	33.0	29	25.7	
University	104	33.3	13	11.5	
Postgraduate	14	4.5	1	0.9	
Missing data *n* = 3	-	-	-	-	
Marital status					
Married	244	78.5	7	6.2	0.045 *
Unmarried	30	9.6	87	77.0	
Devoiced	21	6.8	5	4.4	
Widowed	16	5.1	14	12.4	
Missing data *n* = 4	-	-	-	-	
Self-reported health status					
Poor	127	41	40	35.4	0.300
Good	183	59	73	64.6	
Missing data *n* = 5	-	-	-	-	
Health anxiety					
Anxious	121	38.8	56		0.046 *
Not anxious	191	61.2	57		
Missing data *n* = 3	-	-	-	-	

* *p* Value based on chi-square statistics for categoric variables to compare willingness to continue to join future health screening groups and no willingness group, and not statistically significant at 0.05 level.

**Table 3 ijerph-16-01645-t003:** Influence of socio-demographic characteristics on willingness to participate in health screening.

Variable	Odds Ratio	95%(CI)	*p*-Value
Age (year)			
30–39	-	-	-
40–49	2.370	0.493–11.387	0.281
50–59	0.423	0.136–1.317	0.138
60–69	0.387	0.127–1.179	0.095
70+	0.209	0.068–0.645	0.006 *
Missing data *n* = 3	-	-	-
Education level			
Illiteracy	-	-	-
Elementary	1.025	0.465–2.259	0.951
Junior high school	1.342	0.565–3.188	0.505
Senior high school	3.019	1.402–6.499	0.005 *
University	6.800	2.859–16.171	<0.001 *
Postgraduate	11.900	1.415–100.066	0.023 *
Missing data *n* = 3	-	-	-
Marital status			
Unmarried	-	-	-
Married	0.654	0.277–1.544	0.333
Devoiced	0.980	0.274–3.510	0.975
Widowed	0.267	0.090–0.794	0.018 *
Miss data *n* = 4			
Health anxiety			
Anxious	-	-	-
Not anxious	1.551	1.005–2.392	0.047 *
Missing data *n* = 3	-	-	-

* *p* Value is based on comparison with influence of socio-demographic characteristics and not statistically significant at 0.05 level.

**Table 4 ijerph-16-01645-t004:** Key factors which influence willingness to join health screening services (*N* = 428).

Factors	Incentives to Participate in Health Screening						Willingness to Participate in Health Screening Again			
	Strongly Disagree	Disagree	Neutral	Agree	Strongly Agree	Missing Data	Variable	aOR	95%CI	*p*-Value
My willingness to participate was influenced by the design and quality of the health services.	9 2.1%	25 5.8%	52 12.1%	195 45.6%	145 33.9%	2 0.5%	Neutral or disagree Agree	- 0.826	- 0.456–1.497	- 0.529
The concern of my friends and family motivated me to participate.	14 3.3%	19 4.4%	80 18.7%	165 38.6%	146 34.1%	4 0.9%	Neutral or disagree Agree	- 0.493	- 0.276–0.881	- 0.017 *
I was motivated to participate due to the convenient location of the health screening services.	6 1.4%	3 0.7%	9 2.1%	151 35.3%	256 59.8%	3 0.7%	Neutral or disagree Agree	- 0.856	- 0.264–2.774	- 0.795
I was motivated to participate in health screening because I wished to better understand my health status.	2 0.5%	2 0.5%	9 2.1%	166 38.8%	246 57.4%	3 0.7%	Neutral or disagree Agree	- 0.635	- 0.177–2.282	- 0.487
The free gifts offered to participants were the main reason that I chose to participate.	4 0.9%	14 3.3%	99 23.1%	172 40.2%	130 30.4%	9 2.1%	Neutral or disagree Agree	- 0.509	- 0.321–0.807	- 0.004 *
I feel quite healthy at present and wish to maintain my good health.	6 1.4%	84 19.6%	77 18.0%	180 42.1%	78 18.2%	3 0.7%	Neutral or disagree Agree	- 0.713	- 0.441–1.154	- 0.169

Note: aOR, odds ratio adjusted for education level, sex, and marital status; * *p*-Value < 0.05.

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
