# Peer review of "Primary Drivers of Willingness to Continue to Participate in Community-Based Health Screening for Chronic Diseases"

_ijerph, 2019, doi:10.3390/ijerph16091645_

Round 1
Reviewer 1 Report
This study aims is to identify the sociodemographic factors which influence health screening participation rates and identify the primary drivers of patient willingness to undergo health screening for chronic diseases. This article and the objective are very general and provides little new information on patient willingness.
P2l73 this study was integrated into a community health screening service, we don’t know what is it exactly and which screening were usually realized in these structure, I suppose that there is blood pressure but also cancer screening? Can we really place these two screening on the same level concerning patient willingness
P2l75 the study concerns four districts: the Anle District of Keelung City, and the Ruifang, Gongliao, and Wanli Districts of 75 New Taipei City.The paper could be much more original if it's districts were characterized and if the differences between these districts were studied. A link could be made between the characteristics of the districts and the differences observed in the patient willingness.
P2l87 the study participants receive free health screening, which evaluated indicators pertaining to body mass index, blood pressure, cholesterol, blood glucose level, and presence/absence of helicobacter pylori.But there is maybe other screening like cancer screening as mentioned above.
P2l90 the questionnaire excludes disadvantaged people and / or those with reading and writing difficulties. This limit is not mentioned in the discussion.
P3l104 there is a strong link between screening adherence and socio-economic status. Complementary variables such as the employment held could help to assess the individual socio-economic level of the participants.
In general, the results would be higher if they were presented less globally, by age class, by type of screening. You mentioned it elsewhere in discussion (p9l207) With the current presentation, there is no point of view of original results. You could gain in originality by studying the district effect.
Author Response
Thank you for your suggestions, the response can be seen on the file.
please kindly find the attached
Angela

Reviewer 2 Report
This is an interesting manuscript about the willingness to participate in health screenings. Although it is a well-written manuscript addressing a relevant subject, I have my concerns about the used method to study this subject.
Major comments:
1. When studying willingness to participate, I expect a population with persons who do and do not want to participate. In this study all persons already participated and were asked whether they want to participate in future screenings. In other words, the authors did not measure willingness to participate, but willingness to participate again. This should be mentioned in the title, introduction section and discussion section.
2. Could you provide more information about the invitation of participants? In my country, we are happy to get a response rate of 40% and I am wondering how it is possible to get a response rate of 97%. It would also be interesting to know the characteristics about the non-respondents.
3. In the introduction the authors are focussing on chronic diseases, however based on the content of the health check, this should be limited by cardiometabolic / cardiovascular diseases.
4. The discussion section should include information about the setting. For instance, response can be related to the health care professional who invites the person for a health check.
Other comments:
- Sentence 73-78: this should be described in the methods section
- Our research group did a similar study about willingness to participate (Petter etal BMC Public Health. 2015 Jan 31;15:44) and we wrote a review about non-response in prevention programs (Koopmans etal BMC Public Health. 2012 Oct 9;12:856). These studies can be helpful for the introduction and discussion section.
Author Response

(The authors gave the same response as above.)

Round 2
Reviewer 2 Report
I am happy with the adjustments made by the authors. I have no further comments.